# Dental Implant Nano-Engineering: Advances, Limitations and Future Directions

**DOI:** 10.3390/nano11102489

**Published:** 2021-09-24

**Authors:** Yifan Zhang, Karan Gulati, Ze Li, Ping Di, Yan Liu

**Affiliations:** 1Department of Oral Implantology, Peking University School and Hospital of Stomatology & National Clinical Research Center for Oral Diseases & National Engineering Laboratory for Digital and Material Technology of Stomatology & Beijing Key Laboratory of Digital Stomatology, Beijing 100081, China; zyfbjmu2012@163.com; 2School of Dentistry, The University of Queensland, Herston, QLD 4006, Australia; k.gulati@uq.edu.au; 3School of Stomatology, Chongqing Medical University, Chongqing 400016, China; lizhecqmu@126.com; 4Laboratory of Biomimetic Nanomaterials, Department of Orthodontics, Peking University School and Hospital of Stomatology & National Clinical Research Center for Oral Diseases & National Engineering Laboratory for Digital and Material Technology of Stomatology & Beijing Key Laboratory of Digital Stomatology, Beijing 100081, China

**Keywords:** dental implants, osseointegration, TiO_2_ nanotubes, surface modification, nanoparticles, antibacterial

## Abstract

Titanium (Ti) and its alloys offer favorable biocompatibility, mechanical properties and corrosion resistance, which makes them an ideal material choice for dental implants. However, the long-term success of Ti-based dental implants may be challenged due to implant-related infections and inadequate osseointegration. With the development of nanotechnology, nanoscale modifications and the application of nanomaterials have become key areas of focus for research on dental implants. Surface modifications and the use of various coatings, as well as the development of the controlled release of antibiotics or proteins, have improved the osseointegration and soft-tissue integration of dental implants, as well as their antibacterial and immunomodulatory functions. This review introduces recent nano-engineering technologies and materials used in topographical modifications and surface coatings of Ti-based dental implants. These advances are discussed and detailed, including an evaluation of the evidence of their biocompatibility, toxicity, antimicrobial activities and in-vivo performances. The comparison between these attempts at nano-engineering reveals that there are still research gaps that must be addressed towards their clinical translation. For instance, customized three-dimensional printing technology and stimuli-responsive, multi-functional and time-programmable implant surfaces holds great promise to advance this field. Furthermore, long-term in vivo studies under physiological conditions are required to ensure the clinical application of nanomaterial-modified dental implants.

## 1. Introduction

### 1.1. Dental Implants: History, Survival Rates and Related Complications

In the 1960s, the first preclinical and clinical studies revealed that implants made of commercially pure titanium (Ti) could achieve anchorage in bone, which shifted the paradigm in implant dentistry [1]. Direct bone-to-implant contact, known as osseointegration, formed the foundation of oral implantology [2]. In the next two decades, other materials and different shapes of implants were clinically tested, such as ceramic implants made of aluminum oxide [3], non-threaded implants with a Ti plasma-sprayed surface [4], and Ti-aluminum-vanadium implants [5]. By the end of the 1980s, commercially pure Ti became the preferred material choice of implants [6]. In the 1990s, research findings reported that significantly stronger bone response and higher bone-to-implant contact were achieved in moderately rough or microrough implant surfaces [7]. Next, sandblasted and acid-etched surfaces, as well as microporous surfaces produced by anodic oxidation, were marketed [8,9]. In the past 10 years, zirconium dioxide implants showed comparable preclinical and clinical outcomes as those of moderately rough Ti implants [10]. Currently, microrough implant surfaces are the ‘*gold standard*’ in implant dentistry.

Dental implant treatment is highly predictable, with a survival rate of around 95% according to 10-year clinical observations [11,12,13]. Despite the favorable clinical results, there are still implant-related mechanical, biological and functional complications [14,15]. One major complication is peri-implantitis, which can cause bone loss around the implant, eventually leading to implant failure. According to several reviews, more than 20% of patients and 10% of implants will be affected by peri-implantitis 5–10 years after implantation [16,17]. 

### 1.2. Current Ti Surfaces and Their Physicochemical Modifications

Various implant characteristics influence the osseointegration of dental implants, such as implant geometry (parallel-walled, root-form, conical), thread design and implant-abutment connection (trichannel, external hexagon, internal conical hexagon), as well as implant surfaces [18,19]. Among them, the surface characteristics of dental implants are important determinants of short-term and long-term clinical performance [20,21,22]. Various attempts have been made to optimize implants’ bioactivity by increasing their surface roughness and performing physicochemical modifications, which have reduced the incidence of implant failures and peri-implantitis. 

In the 1980s, the majority of marketed implants featured turned or machined surfaces, with an estimated average roughness (Ra) of 0.5 μm to 0.8 μm. Later, a much rougher surface, Ti plasma sprayed surface (TPS), as well as surfaces coated with hydroxyapatite (HAp) and calcium phosphate (CaP), emerged, with an Ra value of >2 μm [14]. However, these TPS implants coated with HAp soon disappeared from the market, owing to the delamination of the HAp-coating, which can cause severe marginal bone resorption and even implant failure. Next, moderately rough surfaces manufactured by blasting, etching, and oxidation techniques were introduced to the market during the 1990s and early 2000s. One of the most successful surfaces in current clinical implant dentistry is the sandblasted, large-grit, acid-etched (or SLA) surface. Smooth titanium implant surfaces are formed into a primary mechanical cavity of about 200 μm using sand blasting technology, and subsequently cleaned by acid etching to form a secondary cavity of 20μm, resulting in a multi-level rough implant surface, which is conducive to bone bonding. It is worth noting that the 10-year survival rate of SLA Ti implants was reported to be 95–97% [11,23,24]. As a mainstream dental implant surface treatment technology, SLA surfaces have been frequently applied in clinical practice. 

Another comparable surface is produced by using the anodic oxidation technique, which uses a Ti implant as an anode to form a thickened and roughened TiO_2_ layer upon electrochemical treatment. This surface is characterized as isotropic, with an Ra value between 1 μm and 1.5 μm [25]. A recently published meta-analysis comparing the 10-year clinical outcomes of different dental implant surfaces (machined, blasted, acid-etched, sandblasted and acid-etched, anodized, Ti plasma-sprayed, sintered porous and micro-textured) demonstrated that the anodized implants had the lowest failure rate (1.3%, 0.2–2.4%) and minor peri-implantitis rate (1–2%) [14]. According to this article, in addition to a moderate microroughness that increases surface area and oxide thickness, anodized implants also provide additional adhesion points for proteins and cells, which contributes to the augmentation of osseointegration [26]. 

It is well established that osteogenic cells prefer and respond to microrough Ti surfaces, as compared to the machined surfaces [7,27]. However, additional investigations are needed to find the most optimized implant surface topography (SLA or anodized) that enhances bioactivity and osteogenesis [28]. Currently, both SLA and anodized implants present a suitable topography for clinical use. However, SLA surfaces remain the preferred choice in clinical dentistry, with many manufacturers opting for SLA over anodized implants.

### 1.3. Nano-Scale Modifications and Coatings of Ti Implant Surfaces

While micro-roughness is regarded as the ‘gold standard’ towards establishment of appropriate implant-bone bonding, nano-engineering is emerging as a new platform for further enhancement of the dental implant bioactivity. It has been established by several studies, in both in vitro and in vivo settings, that the nano-scale surface modification of Ti implants offers enhanced bioactivity, outperforming the clinical micro-roughness [29]. To fabricate nano-engineered Ti implant surfaces, various strategies have been employed, including plasma treatment, micro-machining, polishing/grinding, particle blasting, chemical etching and electrochemical anodization [30]. The following summarizes the various techniques utilized in the fabrication nano-engineered dental implants.

#### 1.3.1. Mechanical Modification 

While techniques including grinding, machining, blasting and polishing have been used in the production of rough/smooth surfaces, attrition can be used to produce nano-scale layers in order to improve mechanical characteristics, such as hardness and wettability. Machining, polishing and grid-blasting involve the shaping or removal of material surfaces and have been extensively utilized in the fabrication of controlled micro-scale surface topographies on dental implants. Further, machining can result in the deformations of crystalline grains, which increases the surface hardness. The polishing of implant surfaces has also been utilized to obtain smoother finishes. The blasting of abrasive particles against the implant surface can enhance surface reactivity. It is noteworthy that micro-scale surface texturing may be inadequate for the early establishment and subsequent maintenance of osseointegration, especially in compromised conditions. Attrition can enable nanoscale surfaces on implants, which can improve tensile properties, surface hardness and hydrophilicity. 

#### 1.3.2. Chemical Modification 

Changing surface chemistry enables the alteration of topography, as well as the incorporation of chemical moieties that can augment bioactivity and corrosion resistance and offer surface decontamination. A simple acid or alkaline immersion can impart unique surface chemistries/topographies, which have shown promising outcomes. A few tens of nanometers or few micrometers of surface oxide have enhanced the osteogenic potential of implants. Similarly, sol-gel and chemical vapour deposition (CVD) have also been utilized to promote the bioactivity of conventional dental implants. 

#### 1.3.3. Physical Modification 

Processes such as thermal treatment, physical vapour deposition (PVD), ion implantation and plasma treatments are included in the physical modification of implants. The involvement of either thermal, kinetic or electrical energy drives the deposition of specific molecules or ions on the implants’ surface. For instance, thermal or plasma spraying has been used in the coating of hydroxyapatite, calcium silicate, alumina, zirconia and titania on Ti implants, to augment their wear and corrosion resistance and bioactivity. Further, PVD and sputtering also enable favorable biocompatibility and wear/corrosion resistance. Besides, glow discharge plasma can be used not only for surface oxidation, but also for its sterilization.

#### 1.3.4. Electrochemical Modification 

Anodic oxidation enables the growth of 10 nm to 40 μm of TiO_2_ oxide layer and can also allow the adsorption and incorporation of ions from the electrolyte. Through anodic oxidation, controlled topographies can be fabricated on implants, which also offers corrosion resistance and augmented bioactivity. Alternatively, in electrochemical anodization (EA), fluoride and water in electrolytes drive the self-ordering of controlled metal oxide nanostructures when the implant (anode) and counter electrode (cathode) are immersed, and appropriate current/voltage is supplied [31]. 

#### 1.3.5. Biomolecule Modification 

The coating of bioactive molecules, such as collagen or peptides, has been performed on dental implants to enhance bone-implant contact and peri-implant bone formation [32,33]. Additionally, inherently bioactive and antibacterial polymers such as chitosan have also been used in order to modify implant surfaces [34]. Bioactive modifications can induce specific cell and tissue responses, as well as biomimetic precipitation of CaP, via their immersion in simulated body fluid.

## 2. Nanoscale Dental Implant Modifications

### 2.1. Titania Nanotubes 

#### 2.1.1. Fabrication Optimization 

Titania (TiO_2_) nanotubes (TNTs) can be fabricated on Ti or its alloys via electrochemical anodization (EA) [35]. Briefly, EA involves the immersion of a Ti implant as an anode and a bare Ti/Pt electrode (cathode) inside an electrolyte (containing fluoride and water), with the supply of adequate current/voltage [31]. Under controlled and optimized conditions and the attainment of an equilibrium (characterized by metal oxide formation and dissolution), the self-ordering of TiO_2_ nanotubes (like test-tubes, open at the top and closed at the bottom) or nanopores (nanotubes fused together, with no distance between them) on the entire surface of the implant occurs [36]. It is noteworthy that EA represents a cost-effective and scalable Ti implant surface modification strategy. Recent attempts to optimize EA to enable clinical translation include fabrication of controlled nanostructures on clinical dental implants [36], superior mechanical stability (nanopores > nanotubes) [37], and fabrication of dual micro-nano structures [38] by preserving the underlying ‘gold standard’ micro-roughness of dental implants [39]. It is worth noting that EA is a versatile technique that can be used to nano-engineer controlled topographies on various biomedical implants, spanning various metals and alloys, including Ti [40], Ti alloys [41], Zr [42] and Al [43]. A schematic representation of TNTs and their various characteristics and research challenges is shown in Figure 1. 

#### 2.1.2. Osseointegration

Attributed to improved bioactivity and the ability to load and release proteins/growth factors, TNTs are a promising surface modification strategy for orchestrating osteogenesis, as established by various in vivo investigations [29,45]. The incorporation of fluoride ions into TNTs during anodization and the mechanical stimulation of osteoblasts also contribute towards the enhancement of osseointegration [46]. Further, to ensure the successful establishment and maintenance of osseointegration, TNTs on Ti implants have loaded with various orthobiologics, including bone morphogenetic protein-2 (BMP-2) [47], platelet-derived growth factor-BB [48], alendronate [49], ibandronate [50], N-acetyl cysteine (NAC) [51], and parathyroid hormone (PTH) [52]. Lee et al. loaded TNT-modified dental mini-screws with N-acetyl cysteine [NAC, a reactive oxygen species (ROS) scavenger with anti-inflammatory and osteogenic properties], implanted them in rat mandibles in vivo and, at 4 weeks, observed significantly enhanced osteointegration at the NAC–TNT sites [51]. In another study, machined dental implant screws were modified with HF etching and EA to fabricate dual micro- and nanotubular structures, which, upon implantation in ovariectomized sheep in vivo for 12 weeks, showed significantly increased pull-out force and bone-implant contact [53]. Further, various nanoparticles, ions or coatings of Sr [54], Ta [55], La [56], and Zn [57] onto/inside TNTs have also shown upregulated osteogenic outcomes. 

It is worth noting that various ions or NPs have exhibited favorable osseointegration through their use in in vitro and in vivo investigations; however, these may illicit immunotoxic reactions in a dose-dependent manner and remain the subject of active research. Further, with respect to bone-forming proteins, future investigations into the estimation of the local need for bioactive agents and the evaluation of their release inside the bone micro-environment are needed.

#### 2.1.3. Soft-Tissue Integration (STI)

Studies relating to the use of TNTs for enhancing STI for dental implants are very limited, as reviewed elsewhere [58]. Recently, Gulati et al. reported the enhanced proliferation and adhesion of human gingival fibroblasts (HGFs) on dual-micro-nano anisotropic TiO_2_ nanopores [38]. Further, beginning at 1 day of culture, the HGFs started to align parallel to the nanopores; and the gene expression analysis (type I collagen, type III collagen and integrin β1) indicated a wound-healing profile that promoted substrate–cell and cell–cell interactions [59]. Further, anodization combined with heat treatment has also been used to upregulate fibroblast activity. Briefly, the proliferation and adhesion of gingival epithelial cells were enhanced on heat-treated anodized Ti surfaces, which was attributed to hydrothermal treatment precipitation of hydroxyapatite crystals [60]. Alternatively, hydrothermally treated TNTs have been reported to upregulate the integrin α5 and β4 expressions of gingival epithelial cells [61], the adhesion of murine fibroblast-like NIH/3T3 cells and the expression of adhesion kinase [62], as compared to unmodified TNTs.

The biofunctionalization of TNTs has also been explored in order to enhance the functions of fibroblasts and epithelial cells towards augmenting STI. For instance, Xu et al. reported that the inhibition of human gingival epithelial cells on TNTs was reversed when the electrochemical deposition of CaP was performed on TNTs, which was attributed to the local elution of Ca and P ions [63]. Next, Liu et al. investigated the influence of bovine serum albumin (BSA) loading inside TNTs on HGF functions [64]. Unmodified TNTs promoted early HGF adhesion and COL-1 secretion; however, BSA-TNTs enhanced early HGF adhesion, while suppressing late proliferation and COL-1 secretion. It is interesting that contradictory behaviors among bioactive coatings on TNTs have been reported and further in-depth investigation into the influence of these modifications on the STI performance is needed. Furthermore, the local elution of fibroblast growth factor-2 (FGF-2, immobilized on Ag nanoparticles) from TNTs effectively enhanced the proliferation, adhesion and extra-cellular matrix formation in the cultured HGFs [65]. Augmented proliferation, adhesion, and expression of VEGF and LAMA1 genes in vitro was observed, which were pronounced after the loading of 500 ng/mL of FGF-2.

#### 2.1.4. Antibacterial Functions

The local release of therapeutics from TNTs has been widely explored towards optimizing the loading and local elution of potent antibacterial agents [30]. It is noteworthy that within minutes of implantation, saliva proteins adhere to the dental implant, forming a pellicle, and early colonizers such as *Streptococci* adhere to these pellicles within 48h [66]. This can be followed by secondary colonizers, including *Fusobacterium nucleatum*, *Aggregatibacter actinomycetemcomitans* and *Porphyromonas gingivalis* [67]. These bacteria can further lead to peri-implantitis [68]. Once a biofilm is established, the routine administration of antibiotics is insufficient and, hence, local therapy using dental implants has been proposed. Further, TNTs can enhance bacterial adhesion due to their nano-scale roughness, increased number of dead bacteria and amorphous nature. Hence, the synergistic antibacterial functions of TNT-modified dental implants are needed to prevent bacterial colonization and implant failure. Further, the size and crystal structure of TNTs influences bacterial adhesion properties. Ercan et al. investigated the influence of the size and the heat treatment of TNTs on their antibacterial effect and reported that heat-treated and 80 nm diameter TNTs exhibit strong antibacterial effects [69]. Similarly, when comparing 15, 50 and 100 nm diameter TNTs, the lowest number of adherent bacteria were reported on the smallest-diameter TNTs [70]. Further, annealed TNTs show the best bactericidal response, as reported by Mazare et al. [71] and Podporska-Carroll et al. [72]. 

Various commonly prescribed antibiotics including Gentamicin [73], Vancomycin [74], Minocycline, Amoxicillin, Cephalothin [75], Cefuroxime [76] and Cecropin B [77] have been incorporated inside TNT-modified Ti implants to enable local antibacterial functions. Further, to target methicillin-resistant *Staphylococcus aureus* (MRSA), antimicrobial peptides (AMPs) such as HHC-36 have been loaded inside TNTs to achieve a bactericidal effect of almost 99.9% against MRSA [78]. Biopolymer coatings have also been applied to antibiotic-loaded TNTs to: (a) control drug release, (b) promote bioactivity, and (c) harness the inherent antibacterial property of biopolymers in order to provide long-term antibacterial functions. As a result, bare/drug-loaded TNTs have been modified with chitosan [79], polydopamine [80], silk fibroin [81] and PLGA (poly(lactic-co-glycolic acid)), which exhibited synergistic bioactivity and antibacterial enhancements. In addition, various antibacterial ions and nanoparticles (NPs), such as Ag [82], Au [83], Cu [84,85], B, P, Ca [86], Ga [87], Mg [88], ZnO [89], etc., have also been immobilized on or incorporated inside TNTs, with or without the use of hydroxyapatite or biopolymers, using techniques such as micro-arc oxidation, chemical reduction, photo-irradiation, spin-coating, and sputtering.

Multiple synergistic therapies, including osseointegration, immunomodulation, soft-tissue integration and antibacterial functions can also be enabled using nano-engineered Ti with TNTs. For instance, TNTs modified by Ag via plasma immersion ion implantation (PIII) showed excellent antibacterial effects against *P. gingivalis* and *A. actinomycetemcomitans*, while enhancing the bioactivity of epithelial cells and fibroblasts in vitro and reducing inflammatory responses in vivo [90]. Similarly, hydrothermally doped Mg-TNTs exhibited upregulated osteoprogenitor cell adhesion and proliferation (without cytotoxicity) and suppressed osteoclastogenesis, while showing long-lasting antimicrobial effects against methicillin-susceptible *S. aureus* (MSSA), methicillin-resistant *S. aureus* (MRSA) and *E. coli* [88]. 

#### 2.1.5. Immuno-Modulation

The modulation of the host immuno-inflammatory response is crucial to the timely establishment of osseointegration. Hence, attempts have been made to obtain immunomodulatory functions from modified TNTs [44]. These include the influence of physical/chemical characteristics and the local elution of anti-inflammatory drugs from TNTs. The influence of Ti nanotopography on immune cells, including macrophages, monocytes and neutrophils, has supported the attenuation of inflammation [91,92]. Clearly, the presence of nano-scale cues controls macrophage adhesion and inflammatory cytokine production. Similarly, in vitro cultures of such cells on TNTs have also established the influence TNTs nanotopography on immuno-inflammatory responses [38]. 

Smith et al. reported reduced functions (viability, adhesion, proliferation and spreading) of immune cells on TNTs, as compared with bare Ti [93]. Alternatively, other studies have shown enhanced nitric oxide and the absence of foreign-body giant cells on TNTs [93,94]. With respect to the nanotube diameters, inconsistent results have been obtained, with some studies indicating 60–70 nm diameters as the most immuno-compatible [93,95]. Further, Ma et al. compared the functions of monocytes/macrophages on nanotubes and polished Ti, and reported post-attachment stretching inhibition (repulsed adhesion), enhanced M2 phenotype (wound healing) and suppressed M1 phenotype (pro-inflammatory) polarization for TNTs anodized at 5V [96]. Furthermore, to understand the mechanism behind selective immunomodulation due to TNTs, Neacsu et al. reported that this effect is attributed to the suppression of the phosphorylation of MAPK (mitogen-activated protein kinase) signaling molecules (p38, ERK1/2, and JNK) on TNTs [97]. More recently, using 50 and 70 nm diameter anodized anisotropic TiO_2_ nanopores, we showed that macrophage proliferation was significantly reduced on the 70 nm nanopores [38]. Further, the spread of macrophage on nanopores indicated an oval morphology, which was suggestive of an inactivated state. 

The local elution of potent drugs, such as non-steroidal anti-inflammatory drugs (NSAIDs), bypasses the limitations associated with systemic administration (delayed bone healing and toxicity). These drugs have been loaded inside TNTs for the purpose of local release. Briefly, Ibuprofen [98], Indomethacin [99], Dexamethasone [100], Aspirin [101], Sodium naproxen [102], Quercetin [103], Enrofloxacin [104], Propolis [105] and immunomodulatory cytokines [106] have been successfully loaded and locally eluted from TNTs in vitro. Further, to achieve substantial loading and the delayed/controlled release of anti-inflammatory drugs, approaches including biopolymer coating on drug-loaded TNTs [107,108], polymeric micelle encapsulation of drugs prior to loading [109], the periodic tailoring of TNTs [110], the chemical intercalation of drugs inside TNTs [111] and trigger-based release [112,113] have been reported for TNT-based Ti implants. Additionally, metal ions and nanoparticles (NPs), including Au [83], Ag [114] and Zn [115] have also been incorporated on/inside TNTs to impart synergistic immunomodulatory functions with antibacterial or osteogenic activity. More recently, super-hydrophilic TNTs were fabricated via anodization and hydrogenation, and significantly reduced macrophage proliferation; upregulated M2 and downregulated M1 surface markers were exhibited on the modified TNTs, translating into effective immunomodulation and wound healing functionality [116]. It is also noteworthy that various in vivo tests of TNT modifications intended for use in various therapies, including antibacterial [30], osteogenic [29] or anti-cancer [117] applications, have established the immuno-compatibility of TNTs. 

### 2.2. Nanoparticles

NPs can enable multiple therapies at the surface of dental implants, including antibiofouling, osseo- and soft-tissue integration and immunomodulation [118,119]. While NPs have been utilized towards controlled therapies for periodontal, orthodontic, endodontic and restorative treatments, this section will primarily focus on the uses of NP-modified Ti dental implants in implant-based local therapy [120]. As reported in the previous section, NP-doped TNTs have also been widely explored in the context of the controlled release of NPs, which aims to strike a balance between therapy and toxicity.

#### 2.2.1. Silver

Ag NPs are one of the most widely used dentistry restoration and dental implant doping choices due to their outstanding antimicrobial properties [121]. Ag adheres to the bacterial cell wall and the cytoplasmic membrane electrostatically, which causes structural disruption [120]. This results in extensive damage to bacterial DNA, proteins and lipids, resulting in the inhibition of bacterial growth/viability and effective bactericidal action. Besides, Ag NPs can also stimulate osteogenesis and soft-tissue integration, making them an ideal choice for dental implant surface modification [122]. For instance, dental abutments modified with Ag NP suspension prevented *C. albicans* contamination, in comparison with the controls of unmodified abutments [123]. Further, citrate-capped Ag NPs offered bactericidal effects against *S. aureus* and *P. aeruginosa* [124]. Ti implants deposited with Ag NPs using anodic spark deposition have also been co-doped with Si, Ca, P and Na ions, to offer synergistic antibacterial (*S. epidermidis*, *S. mutans* and *E. coli*) and osteogenic (human osteoblast-like cells, SAOS-2) functions [125]. Similarly, to confirm that the used dosage of Ag NPs is safe, a culture of HGFs on Ag NPs/Ti was performed in vitro and the results confirmed no adverse effects [126]. Further, Ag NPs have also been immobilized on Ti implants pre-modified with hydroxyapatite [127], hydrogen titanate [128], chitosan/hyaluronic acid multilayer [129], nanoporous silica coatings [130], Pt and Au [131], and sandblasting and acid-etching [132] in order to achieve superior antibacterial and bioactivity effects. However, while Ag NPs offer effective antimicrobial action, they may cause cytotoxicity via the release of free Ag+ ions, ROS production, transport across blood-brain-barrier, and inflammation [120]. In a manner that is also applicable to other NPs discussed below, the toxicity of NPs depends on their chemical composition, surface charge, size and shape [133]

#### 2.2.2. Zinc

Like Ag NPs, Zn/ZnO NPs are not only antimicrobial but also osteogenic, hence their use in the modification of dental implants [118]. Zn is an essential element in all biological tissues and offers antibacterial effects against a wide range of microbes; however, its aggregation can cause cytotoxicity in mammalian cells [134]. To demonstrate its effectiveness against oral biofilms, Kulshrestha et al. reported that graphene/zinc oxide nanocomposite showed a significant reduction in biofilm formation [135]. Further, Hu et al. incorporated Zn into TiO_2_ coatings on Ti implants through plasma electrolytic oxidation and observed superior bactericidal and bone-forming effects [136]. In 2017, Li et al. synthesized N-halamine labeled Silica/ZnO hybrid nanoparticles to functionalize Ti implants to enable antibacterial functions [137]. The hybrid NP-modified Ti exhibited excellent antibacterial activity against *P. aeruginosa*, *E. coli* and *S. aureus*, without any cytotoxicity against MC3T3-E1 preosteoblast in vitro. Recently, selective laser-melted porous Ti was biofunctionalized using Ag and Zn NPs via plasma electrolytic oxidation and tested against methicillin-resistant *Staphylococcus aureus* (MRSA) [138]. The results confirmed that 75% Ag and 25% Zn fully eradicated both adherent and planktonic bacteria in vitro and ex vivo. Further, Zn-modified Ti (0% Ag) enhanced the metabolic activity of preosteoblasts, indicating its suitability for dual osteogenic and antibacterial implant modification. Further, it is worth noting that ZnO NPs may cause cell apoptosis or necrosis and DNA damage [139] 

#### 2.2.3. Copper

CuO NPs offer advantages over Ag NPs, including cost-effectiveness, chemical stability and ease of combining with polymers, which makes them an attractive choice for biomaterial applications [140]. Further, Cu NPs have antibacterial, osteogenic and angiogenic properties [141], and have been applied towards the enhancement of both the bioactivity and the antimicrobial properties of Ti dental implants [142]. More recently, van Hengel et al. incorporated varying amounts of Ag and Cu NPs into TiO_2_ coating on additively manufactured Ti–6Al–4V porous implants via plasma electrolytic oxidation [143]. Further, 75% Ag and 25% Cu caused the eradication of all bacteria in a murine femora model ex vivo, while only Cu NP-modified implants (0% Ag) augmented the metabolic activity of pre-osteoblastic MC3T3-E1 cells in vitro. Alternatively, Ti-6Al-7Nb alloy dental implants were coated with Cu NPs and cultured with *P. gingivalis* in vitro, and the findings suggested that Cu NPs can aid in local infection control around implants [144]. In 2020, Xia et al. reported the use of plasma immersion ion implantation and deposition (PIIID) technology to modify Ti implants with C/Cu NPs co-implantation [145]. The modified implants displayed superior mechanical and corrosion resistance properties and enhanced the antibacterial performance of Ti implants (against *S. aureus* and *E. coli*) without causing cytotoxicity (to mouse osteoblast cells) in vitro. In a more dental implant setting, Cu-deposited (micro-/nanoparticles) commercially pure (cp) grade 4 Ti discs (via spark-assisted anodization) were shown to exhibit dose-dependent antibacterial effects against peri-implantitis-associated strain *P. gingivalis* [146]. Similarly, micro-arc oxidation Cu NP-doped TiO_2_ coatings showed excellent antibacterial activity, while augmenting the proliferation and adhesion of osteoblast and endothelial cells in vitro [85]. The interaction of Cu NPs with microbes and the bioactivity and toxicity evaluations of Cu NPs can be found elsewhere [147].

#### 2.2.4. Zirconia

Zirconium (Zr) and zirconia (ZrO_2_) are rising as dental implant material choices due to their biocompatibility, corrosion resistance and superior mechanical properties [42]. It is established that Zr^4+^ ions can interact with negatively charged bacterial membranes and cause cell damage and death [148]. Furthermore, Zr-based implants have been electrochemically anodized in order to fabricate controlled ZrO_2_ nanostructures, including nanotubes and nanopores, which can augment implant bioactivity due to their nanoscale roughness [42,149,150]. For instance, anodized Zr cylinders were placed in rat femur osteotomy models in vivo, and accelerated bone formation was obtained, in comparison with the controls of unmodified Zr [151]. Further, Indira et al. reported the dip coating of Zr ions into anodized TNTs to form ZrTiO_4_ over the nanotubes, which exhibited enhanced bioactivity (HAp formation in Hank’s solution in vitro) and corrosion resistance [152]. Similarly, the application of a Zr film on a TiNi alloy via plasma immersion ion implantation and deposition (PIIID) augmented its corrosion resistance [153]. Nanotube formation has also been extended to TiZr alloys. For instance, Grigorescu et al. used two-step EA to fabricate nanotubes of varied diameters and observed an increase in hydrophilicity with reduction in diameter [154]. Further, the smallest nanotube diameters exhibited the highest antibacterial effects against *E. coli*. While ZrO_2_/Zr is extensively used as a dental implant material, the leaching of Zr NPs may initiate cytotoxicity. For instance, the application of both Zr and TiO_2_ NPs in a dose-dependent fashion could lead to osteoblast morphology changes and apoptosis, affecting both osteoblast differentiation and osteogenesis at high dosages [155].

#### 2.2.5. Silica

Si/SiO_2_ NPs have been utilized in biomedical applications, including biosensing and drug delivery [156]. In dentistry, Si NPs have been used as dental filler, for tooth polishing and in hypersensitivity treatments [157]. Varied concentrations of SiO_2_ NPs within HAp fabricated on Ti hydrothermally were analyzed for bioactivity and cytotoxicity [158]. The results confirmed homogenous distribution of SiO_2_ NPs on hexagonal HAp crystals and favourable biocompatibility with human osteoblast-like cells in vitro. Furthermore, in order to achieve superior bioactivity, a protein-based Si NP coating (via the genetic fusion of recombinant MAP with the R5 peptide derived from a marine diatom *Cylindrotheca fusiformis*) was performed on Ti implants to explore their osteogenic potential [159]. Briefly, the assembly of Si NPs augmented the in vitro osteogenic cellular behaviors of preosteoblasts and bone tissue formation in vivo (calvarial defect model). Further, Si NP coatings were performed on Ti-based implants to enable the local elution of potent therapeutics in order to achieve antibacterial [160] and osseointegration [161] functions. For example, the enhancement of osteogenesis via immunomodulation by Si NP-doped chitosan-modified TNTs has been reported [162]. Furthermore, 100 nm mesoporous Si NPs were loaded with dexamethasone (an anti-inflammatory drug) in order to achieve its local elution, which demonstrated favourable macrophage cytocompatibility. Additionally, local release of dexamethasone modulated M2 macrophage polarization which supported osteogenesis. Immuno-toxicity evaluations of Si NPs have been reviewed elsewhere [163].

### 2.3. Hydroxyapatite

HAp is biocompatible, non-toxic and non-immunogenic, and has been widely used as a coating material in the modification of dental implants [164]. In the late 1980s, plasma spray coating of HAp became obsolete due to the delamination of the HAp-coat, which could cause severe marginal bone resorption and incompatibility with antibiotic incorporation [165,166]. Later, some alternative coating techniques, such as electrochemical deposition [167], electrophoretic deposition and electrospray deposition [168] made it possible to combine HAp coatings with antibiotics to achieve both enhanced bioactivity and antibacterial effects. Moreover, due to their special crystalline structure and positive-charged surface, the ability of substituted HAp to immobilize proteins and growth factors through noncovalent interactions has offered new possibilities for the preparation of hybrid coatings that accelerate bone healing [169]. 

Geuli et al. reported the use of drug-loaded HAp nanoparticles on Ti implants through single-step electrophoretic deposition. The release profiles of the gentamicin sulfate (Gs)-HAp and ciprofloxacin (Cip)-HAp coatings demonstrated a prolonged release of up to 10 and 25 days, respectively. In vitro antibacterial tests of the Cip and Gs-HAp coatings showed the efficient inhibition of *P. aeruginosa* [170]. Liu et al. applied a nano-silver-loaded HAp (Ag-HAp) nanocomposite coating to a Ti6Al4V surface by laser melting. They found that the coating containing 2% Ag showed excellent biocompatibility and antibacterial ability, which was conducive to the deposition of apatite on the implant’s surface [171]. Further, Zhao et al. compared the application of magnesium (Mg)-substituted and pure HAp coatings in the osseointegration of dental implants in vitro and in vivo [172]. They observed increased cell proliferation, higher alkaline phosphatase activity and enhanced osteocalcin production in the Mg-HAp group in vitro. In vivo testing using a rabbit femur model revealed a slightly higher bone implant contact for the Mg-Hap-coated implants at 2 weeks post-implantation, whereas no significant differences were seen after 4 and 8 weeks. Recently, Vu et al. coated Ti and Ti6Al4V implants with a ternary dopant coating, which used commercial HA powder doping with 0.25 wt% ZnO to induce osteogenesis, 0.5 wt% SiO_2_ to induce angiogenesis, and 2 wt% Ag_2_O to control infection [173]. The Zn/Si/Ag-HAp coatings resulted in better antibacterial properties in vitro against *E. coli* and *S. aureus*. Meanwhile, the Zn/Si/Ag-HAp implants with higher shear modulus augmented bone mineralization and total bone formation compared to pure HAp implants in rats by week 5, while no evidence of angiogenesis or antibacterial properties, as demonstrated in vivo (Figure 2). Other ionic substitutes, such as Si, F^−^, Sr^2+^ have also been utilized in combination with HAp to accelerate bone healing and improve bioactivity [164]. 

### 2.4. Biopolymers

Polymeric layers are a promising strategy for the enhancement of bioactivity and controlled release of potent drugs. The use of biopolymers, such as chitosan, cellulose and silk fibroin-based nanomaterials provide the synthetic implant surface coatings with superior bioactivity and antibacterial functions. A combination of implant surface treatment with polymer-incorporated antibiotics, drugs or biomolecular delivery systems has shown promising results when compared to polymers and drugs alone. Here, we discuss and detail the application of the two most commonly utilized polymers in dental implants.

Chitosan is an inherently antibacterial and non-toxic polysaccharide that is widely applied in wound healing, tissue engineering and drug delivery [174]. Moreover, nanofibrous chitosan provides a more favorable microenvironment for cellular activity than bulk chitosan, which can be attributed to the way its unique morphological characteristics mimic extracellular matrices [175,176]. Benefiting from its positive surface charge, chitosan is also antibacterial and ruptures negatively charged bacterial cells [177]. Many studies have been conducted on the use of chitosan for the fabrication of antibacterial medical implants [178,179,180]. Furthermore, when chitosan is incorporated in the form of nanoparticles on the implant surface, it shows a high loading rate and the capacity for sustained drug release. Chitosan [181], chitosan/gelatin [182], chitosan/alginate [107] and chitosan/graphene oxide [183] have also been utilized in the coating of implants.

Song et al. used chitosan to wrap Semaphorin 3A (Sema 3A), a proven osteoprotection molecule, and to immobilize oxidized Ti surface. A burst release of Sema 3A was maintained for more than 2 weeks [184] (Figure 3a–e). Further, Mattioli-Belmonte et al. prepared a ciprofloxacin-loaded chitosan nanoparticle-based coating on Ti substrates for the in situ release of the antibiotic for post-operative infections. According to the in vitro results, this coating inhibited the growth of *Staphylococci aureus* and did not impair the viability, adhesion or expression of MG63 osteoblast-like cells [185]. Ma et al. applied chitosan-gelatin (CS/G) coatings to a Ti surface and evaluated its biological *performance* in vitro and in vivo [186]. The CS/G coatings supported MC3T3-E1 cell attachment, migration and proliferation. In addition, Micro-CT and histomorphometrical analysis revealed new bone formation around CS/G implants at 8 and 12 weeks, while the majority of the coatings were degraded at 12 weeks (Figure 3f,g). 

A recent review focused on the toxicity/safety concerns in zebrafish models, and described the toxicity of different chitosan nanocomposites [187]. According to Hu et al., 200 nm chitosan nanoparticles (Ch NPs) were able to cause 100% mortality to the embryos and severe teratogenic deformities at 40 mg/L, compared to the 340 nm particles [188]. By contrast, both Wang et al. [189] using 200 mg/L Ch NPs, and Abou-Saleh et al. [190], using 100–150 nm Ch NPs, failed to induce significant mortality or teratogenic phenotypes, even at 200 mg/L. The contradictory results suggest that more cytotoxicity and toxicity investigations of Ch NPs are required to advance this field.

### 2.5. Carbon Composites

Graphene, obtained through the physicochemical exfoliation of graphite, provides several advantages, such as its low cost and safe preparation. There are several derivative forms of graphene, such as graphene oxide (GO), which is highly oxidative, and reduced GO (rGO), which is prepared via the chemical or thermal reduction of GO. It has been reported that pure graphene shows a certain degree of cytotoxicity [191]. However, whether GO causes cytotoxicity remains controversial, as some studies have shown that GO does not initiate cytotoxicity [192,193,194]. However, others have revealed that micro-sized GO (and not nano-sized GO) can induce high levels of cytotoxicity [195]. It is worth noting that the main difference among all the carbon-based materials is the hybridization type of their carbon atoms [196,197]. In a study by Wang et al., the hybridization type of carbon atoms (sp2 or sp3) was the critical point in determining their biological properties. The larger amount and smaller size of dispersing sp2 domains regulated the behavior of cells by affecting the amount and properties of the adsorbed proteins [198].

Recently, Gu et al., attempted to improve the adhesion strength of graphene on the surface of Ti substrate through a thermal treatment and observed enhanced antibacterial effects (*E. coli* and *S. aureus*), cell adhesion, proliferation and osteogenesis in vitro (human adipose-derived stem cells and human bone marrow mesenchymal stem cells) and in vivo (dorsal subcutaneous area of eight-week-old male BALB/c nude mice) on graphene-coated Ti implant surfaces after dry heating treatment [199]. More recently, Wei et al. synthesized a new Ti biomaterial containing graphene (Ti-0.125G) by using the spark plasma sintering technique. Bioactivity (human gingival fibroblasts) and antimicrobial (*Streptococci mutans*, *Fusobacterium nucleatum*, and *Porphyromonas gingivalis*) findings revealed that graphene modification upregulated both functions [200].

Unlike hydrophobic graphene, GO is a hydrophilic derivative form with the addition of bounded oxygen atoms. Due to the large number of carboxyl and hydroxyl groups containing active functional groups on its surface, it is easy to perform biomaterial functionalization using GO [201,202]. Wang et al. assembled GO coatings on a laser microgroove Ti alloy [203]. The in vitro bioactivity results showed superior adhesion, proliferation, differentiation and osteogenic capability compared with bare Ti implant, due to the wettability and apatite formation induced by the GO coating. Based on the findings of Li et al., the FAK/P38 signaling pathways were proven to be involved in the enhanced osteogenic differentiation of bone marrow mesenchymal stem cells, accompanied by the upregulated expression of focal adhesion (vinculin) on the GO-coated surface [204]. It is noteworthy that GO can cause direct damage to bacterial cell membranes through its sharp structure and its destructive extraction of lipid molecules, together with ROS reactions [205]. Besides, the extensive two-dimensional (2D) honeycomb structure can be loaded with biomolecules or drugs in order to enable local therapy [195]. 

Compared to GO, rGO possesses structural defects that enhance molecular interactions. For instance, Kang et al. fabricated rGO-coated Ti substrates through meniscus-dragging deposition and investigated their biological behaviors [206]. They cultured human mesenchymal stem cells on the Rgo-Ti substrates and found superior bioactivity and osteogenic potential via a cell counting kit-8 assay, an alkaline phosphatase activity assay and alizarin red S staining, suggesting that these graphene derivatives had potent applications in dental implants. Further, Rahnamaee et al. assembled both chitosan nanofibers (CH) and reduced graphene oxide (rGO) onto TNTs [183]. This multifunctional coating offered the synergistic effects of CH and rGO against both long-term and short-term antibacterial activity, promoted osteoblast cell viability, prolonged antibiotic release profile and inhibited bacterial biofilm formation.

Another 2D carbon-based nanomaterial is graphdiyne (GDY), which has been predicted to become the most stable carbon derivative form [207]. Compared with graphene, the particular sp and sp2 hybridized carbon atoms of GDY offer superior electrical conductivity and enhanced catalytic effects, and exhibited enhanced biocompatibility and stability in some in vivo studies [208,209]. Further, Wang et al. successfully assembled GDY onto TiO_2_ to synthesize a TiO_2_/GDY composite by using electrostatic force [210]. Its antibacterial effect against *Methicillin-resistant Staphylococcus aureus* (MRSA) was prolonged with sustained ROS release, which prevented the formation of biofilm. A mouse implant infection model further demonstrated excellent sterilization and bone regeneration effects in vivo (Figure 4). The reviewed research works on various modifications for dental implants with their main advantages and drawbacks are summarized in Table 1.

## 3. Research Challenges

The use of various nano-engineering strategies to enhance the bioactivity and therapeutic performance of dental implants shows great promise; however, many research gaps remain unaddressed with respect to the clinical application of nano-engineered dental implants. Next, we take a close look at the key challenges that must be investigated in order to bridge the gap between nano-engineering dental implant research and its clinical translation.

The key physical, chemical and mechanical characteristics of the implant and its surface modification are crucial towards the understanding and prediction of cell response and therapeutic efficacy [211]. These also include appropriate corrosion resistance and electrochemical stability. Hence, testing under masticatory loading conditions, under varied pH and physiological conditions (matching healthy and compromised conditions, such as infection and inflammation) for extended durations are essential for nano-engineered coatings of implants. Any delamination or release of nanoparticles from implant modifications can initiate a cytotoxic response, and only a few attempts have been made to ensure the successful fabrication of robust nano-engineered coatings on commercial implants with appropriate mechanical stability [36,37].Nano-engineered implants can enable the local elution of potent drugs, proteins or therapeutic nanoparticles/ions. While the concept of local drug release has gained attention, its investigation has largely remained restricted to proof-of-concept in vitro studies or short-term in vivo investigations without mechanical loading. Further, to enable the deep loading of drugs and a controlled initial burst release, drugs have been encapsulated in micelles prior to loading [109], or loaded in TNTs covered with biopolymers [52,108]; however, the release only lasts for a few weeks or 1–2 months. It is noteworthy that therapeutic action may be needed for prolonged periods (several months to years) in order to achieve long-term implant success, specially in compromised conditions.When a drug-releasing implant is placed, several cells ‘*race to invade*’ the site [66], and often the nanotopography is immediately covered with proteins and cells, which may block the open pores [117,212]. This can impact drug release, given that the latter is dependent on a diffusion gradient that is impeded by poor perfusion inside the bone micro-environment. These conditions, especially considering that surgical placement causes trauma, even in healthy patients, may be difficult to approximate in vitro and *in silico* [213]. Hence, the performance of drug-releasing implants must be tested in real traumatized tissue in vivo, based on therapeutic needs identified ex vivo [214].Ideally, the implant surface modification should cater to the *three Is*, *integration* (both osseo- and soft-tissue integration), *inflammation* and *infection*, in order to enable early acceptance and long-term survival. While multi-therapeutic nano-engineered implants have been applied, either by combining various drugs or through the inclusion of biopolymers or metal ions/nanoparticles, their effectiveness in compromised patients conditions including advanced age, diabetes or osteoporosis, has not been investigated. It is worth noting that the success of dental implants is further challenged in these patient conditions. Further, nano-engineering attempts to augment soft-tissue integration in order to form a barrier to the ingress of oral pathogens is not explored adequately.To ensure clinical translation, avoiding the ‘*valley of death*’, nano-engineered implants must survive packaging, handling, implantation and operation inside the dental micro-environment. This also includes optimizations at all stages of product development, from the fabrication of controlled and reproducible nanostructures to bioactivity and local therapy. Further, bioactivity and cytotoxicity evaluations specifically considering initial burst release, the early consumption of drugs and dead bacteria/cells blocking the open pores of TNTs are vital. Additionally, with the use of metals ions and nanoparticles to augment the therapeutic effects of implants, it is important to determine and control their release profile to reduce cytotoxicity.

## 4. Future Perspectives 

The next generation of dental implants will employ optimized nanotopography to simultaneously augment antibacterial and osseointegration functions. The following details the future directions in the domain of nano-engineered dental implants:The integration of new materials and technologies is the key factor in the development of new hybrid dental implants. However, it remains difficult to fabricate uniform nanostructures rapidly and on a large scale. Additive manufacturing or three-dimensional (3D) printing technology may provide customized implants to match patient needs [215]. In 2014, Dong et al. successfully fabricated a novel 3D porous scaffold by mixing anti-tuberculosis bacterium drugs, Poly-DL-lactide and nano-hydroxyapatite via additive manufacturing technology [216]. In the field of orthopedic surgery, the use of 3D printing is increasing and patient-specific implants have been produced to meet the surgical requirements [217,218]. By controlling the shape and porosity using the rapid prototyping method, 3D-printed implants enable rapid bone in-growth and reduce implant stiffness. However, the use of 3D-printed implants is limited due to high costs and time demands. Although it is still in development, 3D printing technology is the most important direction for fabricating future dental implants.Another direction for future dental implants is triggered drug release, whereby the therapeutic payloads are released via an internal or external stimulus, which significantly reduces the initial burst release, ensuring release ‘*on-demand*’ [113]. The triggering mechanisms can be temperature, pH, electric or magnetic fields, or radio or ultrasonic frequencies. Further, future ‘smart’ dental implants could detect/sense the type of cellular attachment or tissue formation around the implant, and switch the release of a drug on or off.

Additive manufacturing, as well as biosensing and triggered drug release techniques are the future of multi-functional and customizable dental implants [219]. 

## 5. Conclusions

The nano-engineering of dental implants has been performed in order to augment the antibacterial and bioactivity performances of conventional implants, improving long-term treatment outcomes. This article reviewed the nano-engineering of Ti-based dental implants and evaluated modifications with titania nanotubes, nanoparticles, biopolymers and carbon-based coatings in terms of biocompatibility, antimicrobial activity, toxicity and in vivo evidence. Various nanoscale dental implant modifications and their key features have been summarized in Table 1. While in vitro and short-term in vivo studies have shown favorable outcomes, long-term in vivo investigations in compromised models (including inflammation and infection), under masticatory loading, are needed to ensure the clinical translation of nano-engineered dental implants. Clearly, the future of dental implants will include customized, patient-specific, nano-engineered implants that enable long-term therapeutic action, while augmenting implant-tissue integration, without initiating any cytotoxicity.

## Figures and Tables

**Figure 1 nanomaterials-11-02489-f001:**
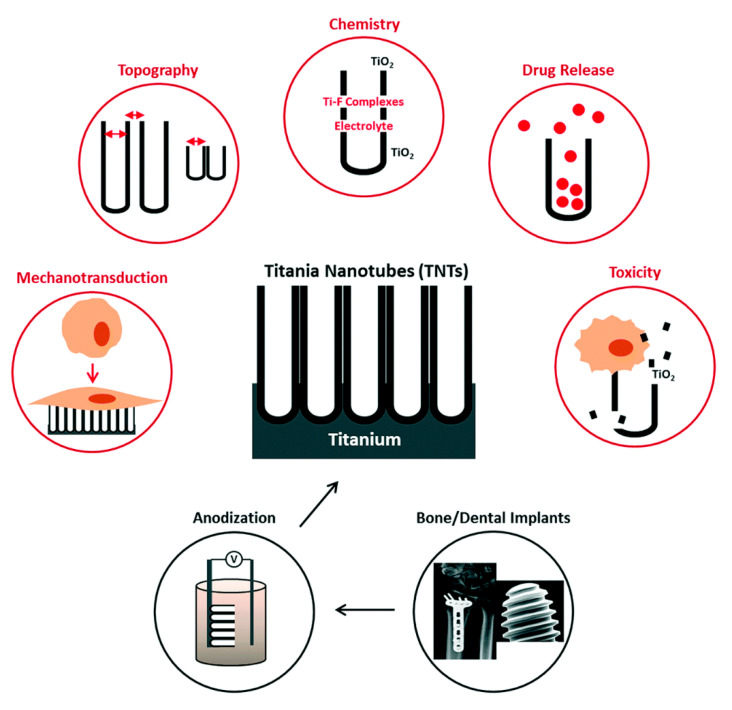
Electrochemically anodized dental implants with titania nanotubes (TNTs) for the purpose of enhanced bioactivity and local therapy. Adapted with permission from [44].

**Figure 2 nanomaterials-11-02489-f002:**
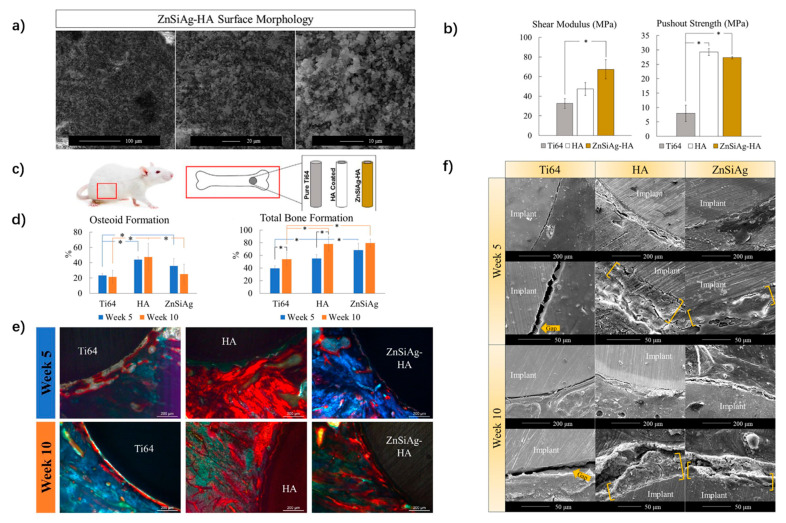
Mechanical and biological properties of Zn/Si/Ag-HAp coating implants. (**a**) Scanning electron microscopy (SEM) showing roughness and porosity of the implant surface. (**b**) Shear modulus and pushout test of implants from harvested femurs. (* *p* < 0.05). (**c**) The in-vivo bilateral model rat’s distal femur. (**d**) Total osteoid formation and bone formation in % around implant at week 5 and week 10. (* *p* < 0.05). (**e**) Modified Masson-Goldner trichrome staining 5 weeks and 10 weeks after implantation. (**f**) SEM images of implant interface for all compositions at week 5 and week 10. Adapted with permission from Vu, et al. Coatings are outlined with yellow brackets [173].

**Figure 3 nanomaterials-11-02489-f003:**
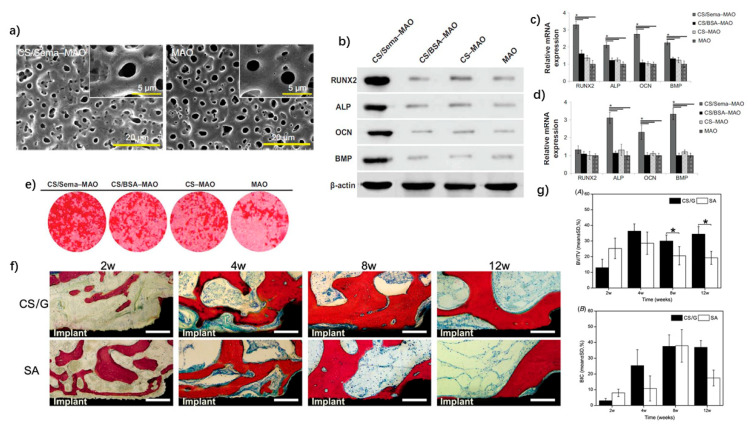
Applications of nano-chitosan in implant surface coatings. (**a**) SEM images showing surface morphology of chitosan–semaphorin 3A-microarc oxidation (CS/Sema-MAO) and MAO. (**b**–**e**) Osteogenic-related gene and protein expression of MG63 cells cultured on CS/Sema-MAO surface and control surface at day 3 (**b**,**c**), day 7 (**d**) and day 21. * *p* < 0.05 vs CS/Sema–MAO, CS–MAO, and MAO. (**e**) Alizarin red study. (**f**) Basic fuchsin and methylene blue staining showing histological appearance around chitosan–gelatin (CS/G) coatings and sandblasted/acid-etched (SA) implants at weeks 2, 4, 8, and 12. (**g**) Histomorphometrical variables evaluating the gap level of CS/G and SA implants within the region of interest: (A) the percentage of bone volume fraction (BV/TV) around the implant within 300 μm; and (B) the ratio of BIC. * *p* < 0.05. Adapted with permission from [184,186].

**Figure 4 nanomaterials-11-02489-f004:**
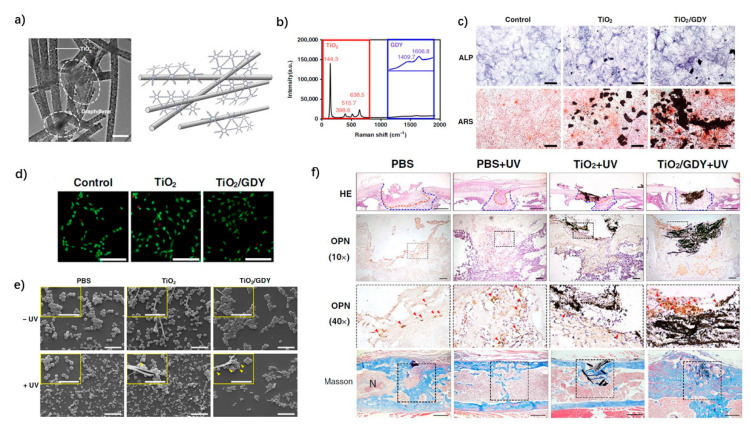
Graphdiyne (GDY)-modified TiO_2_/Ti implants. (**a**) Transmission electron microscopy image of TiO_2_/GDY. (**b**) Raman spectra of TiO_2_/GDY. (**c**) Osteogenic effects of TiO_2_/GDY and TiO_2_ in vitro: alkaline phosphatase activity (upper labeled ALP) and alizarin red S staining (lower-labeled ARS) on day 14. (**d**) Live/dead staining for MC3T3-E1 cells cultured with nanofibers (scale bar = 50 μm). (**e**) SEM images of MRSA biofilms after exposure to different conditions. yellow arrows in the magnified inset images show holes on the bacterial surface; scale bar = 5 μm (upper), 10 μm (below). (**f**) Hematoxylin and Eosin (HE) staining and immunohistochemical staining of infected tissues after 5 days; Masson staining for bone formation after 4 weeks. In vivo implant infection model: femur bone defect with MRSA infection in 8-week-old mouse. Adapted with permission from [210].

**Table 1 nanomaterials-11-02489-t001:** Summary of nanoscale dental implant modifications and their key features.

Implant Modification	Fabrication	Advantages	Drawbacks	Main Reference
TiO_2_ nanotubes	Electrochemical anodization	Enhanced osseointegration		[29,45]
		Soft-tissue integration: enhanced proliferation and adhesion of human gingival fibroblasts		[38]
		Local release of therapeutics		[30]
		Immunomodulatory functions		[94,95]
Ag NPs	Anodic spark deposition	Outstanding antimicrobial properties	Toxicity: via release of free Ag+ ions	[120,121,122]
		Stimulation of osteogenesis and soft-tissue integration		
Zn/ZnO NPs	Plasma electrolytic oxidation	Antibacterial properties Osteogenic effects	Cytotoxicity: ZnO NPs may cause cell apoptosis or necrosis and DNA damage	[118,134,136]
CuO NPs	Plasma electrolytic oxidation	Cost-effectiveness Chemical stability Ease of mixing with polymers	Toxicity	[140,145,147]
	Plasma immersion ion implantation and deposition (PIIID) Micro-arc oxidation	Antibacterial effects Osteogenic properties Angiogenic properties		
ZrO_2_ nanostructures	Electrochemical anodization Plasma immersion ion implantation and deposition (PIIID)	Enhanced bioactivity Corrosion resistance Antibacterial effects	Cytotoxicity: dose-dependent, affecting both osteoblast differentiation and osteogenesis at high dosages	[152,153,155]
Si/SiO_2_ NPs	Hydrothermal method	Biocompatibility with human osteoblast-like cells in vitro Antibacterial properties Immunomodulation		[158,160,162]
Hydroxyapatite	Electrochemical deposition Electrophoretic deposition Electrospray deposition	Biocompatibility Non-toxicity Non-immunogenicity Prolonged drug release		[170,173]
Chitosan	Microarc oxidized and silane glutaraldehyde coupling	Antibacterial properties High loading rate and sustained drug release ability		[185,188]
Carbon composites	Dry heating treatment Meniscus-dragging deposition	Low cost Safer preparation Enhanced antibacterial effects Bioactivity in vitro and in vivo Highly efficient drug loading and therapy	Cytotoxicity: remains controversial	[184,196,200,205,211]

## Data Availability

Not applicable.

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
