# Peer review of "Dental Implant Nano-Engineering: Advances, Limitations and Future Directions"

_nanomaterials, 2021, doi:10.3390/nano11102489_

Round 1
Reviewer 1 Report
The authors summarized the characteristics and research results of implant surface treatment, particularly those using nanostructures such as nanotube or particles, and presented them well. This work would be more important if some parts could be summarized more clearly. Q1: As described in section 1.2, surface roughness of dental implant going advance and suggested anodized implant had the lowest failure rate and side effect. however, in a view of recent industrial trend of surface treatment , anodized implant are not major position of surface treatment. so need to explain more Therefore, please compare the specific parts of various methods of surface treatment and suggest a more mainstream method. Q2: in section 2, add the summarize figure or table including compare each technique quantifiably with specific examples.Author Response
Thanks for carefully reading our manuscript and constructive comments, which have dramatically helped us to improve our manuscript.
Q1: As described in section 1.2, surface roughness of dental implant going advance and suggested anodized implant had the lowest failure rate and side effect. however, in a view of recent industrial trend of surface treatment, anodized implant are not major position of surface treatment. so need to explain more. Therefore, please compare the specific parts of various methods of surface treatment and suggest a more mainstream method.
Our response: Thanks for the Reviewer’s comment. We have added more details on fabrication and clinical evidences both on SLA and anodic oxidation techniques, and demonstrated that currently both SLA and anodized implants are accepted as first choice in clinic (section 1.2, Page 2 Line 56-77.). From the view of microstructure, it is still unknown which surface topography (SLA or anodized) is more proper for osteointegration. While SLA, as one of the mainstream treatment technologies of implant surface, has been frequently tested in clinics for the longest period.
Q2: in section 2, add the summarize figure or table including compare each technique quantifiably with specific examples.
Our response: We sincerely appreciate the Reviewer’s suggestion. We have added Table 1 (Page 13-14) to compare each technique in the revised manuscript.

Reviewer 2 Report
the manuscript give a satisfactory overview of the industrial procedures/ materials for improving the implant surface activity.
Author Response
Thanks for your comments and we greatly appreciate it. We have added one table (Page 13-14) and more detailed introduction in our revised manuscript (section 1.2, Page 2 Line 56-77.). The revised parts are marked using track changes.

Reviewer 3 Report
Interesting summery of the current knowledge .
Author Response
Our response: Thanks for your comments and we greatly appreciate it. We have added one table (Page 13-14) and more detailed introduction in our revised manuscript (section 1.2, Page 2 Line 56-77.). The revised parts are marked using track changes.
